# Current Targets and Bioconjugation Strategies in Photodynamic Diagnosis and Therapy of Cancer

**DOI:** 10.3390/molecules25214964

**Published:** 2020-10-27

**Authors:** Salvador Gomez, Allan Tsung, Zhiwei Hu

**Affiliations:** 1The James-Comprehensive Cancer Center, Division of Surgical Oncology Department of Surgery, College of Medicine, The Ohio State University, 460 W 12th Ave, Columbus, OH 43210, USA; salvador.gomez@osumc.edu (S.G.); allan.tsung@osumc.edu (A.T.); 2College of Medicine, The Ohio State University, 370 W 9th Ave, Columbus, OH 43210, USA

**Keywords:** photodynamic therapy, photodynamic diagnosis, cancer, bioconjugation, tissue factor, genetically encoded fluorescent protein photosensitizer

## Abstract

Photodynamic diagnosis (PDD) and therapy (PDT) are emerging, non/minimally invasive techniques for cancer diagnosis and treatment. Both techniques require a photosensitizer and light to visualize or destroy cancer cells. However, a limitation of conventional, non-targeted PDT is poor selectivity, causing side effects. The bioconjugation of a photosensitizer to a tumor-targeting molecule, such as an antibody or a ligand peptide, is a way to improve selectivity. The bioconjugation strategy can generate a tumor-targeting photosensitizer conjugate specific for cancer cells, or ideally, for multiple tumor compartments to improve selectivity and efficacy, such as cancer stem cells and tumor neovasculature within the tumor microenvironment. If successful, such targeted photosensitizer conjugates can also be used for specific visualization and detection of cancer cells and/or tumor angiogenesis (an early event in tumorigenesis) with the hope of an early diagnosis of cancer. The purpose of this review is to summarize some current promising target molecules, e.g., tissue factor (also known as CD142), and the currently used bioconjugation strategies in PDT and PDD, with a focus on newly developed protein photosensitizers. These are genetically engineered photosensitizers, with the possibility of generating a fusion protein photosensitizer by recombinant DNA technology for both PDT and PDD without the need of chemical conjugation. We believe that providing an overview of promising targets and bioconjugation strategies will aid in driving research in this field forward towards more effective, less toxic, and non- or minimally invasive treatment and diagnosis options for cancer patients.

## 1. Introduction

Photodynamic diagnosis (PDD) and therapy (PDT) are emerging, non/minimally invasive techniques for diagnosis and treatment of cancer [1,2,3]. Besides cancer, PDT has been used to treat age-related macular degeneration [4], microbial infections [5,6], as well as dermatologic, urologic, gynecologic, and oral diseases [7,8,9,10]. Both techniques require a photosensitizer (PS) and light [11]. When activated by a suitable wavelength of light [12], photosensitizers, i.e., light-sensitive molecules, can convert molecular oxygen to generate reactive oxygen species (ROS) to kill the target cells within a short distance to achieve treatment effects in PDT, whereas photosensitizers in PDD can emit fluorescence to illuminate the diseased tissues/cells for visualization of lesions. In PDT, the photosensitizer is delivered to pathologic tissue followed by activation with light exposure. This initiates the formation of highly reactive oxygen species to destroy the diseased lesion [13]. PDD works similarly but does not result in the destruction of the lesion. Rather, current PDD usually uses the fluorescence of a non-targeted photosensitizer without ROS production, for example, 5-Aminolevulinic acid (5-ALA) with blue or white light excitation [14,15], to identify tumor tissue. The fluorescence will thus hopefully mark pathologic tissue under fluorescent light without destroying it. Thus, PDD, when used as an intraoperative fluorescence-guided diagnosis, may allow for more accurate removal of tumors and lower the risk of unnecessary damage to healthy tissue during surgical operation [16,17,18].

PDT is initiated by the presence of light at a specific wavelength, which is dependent on the photosensitizer used. Upon exposure, photosensitizers are excited from the ground state (S_0_) to the excited state (S_1_-S_n_). Then, intersystem crossing (ISC) occurs to bring the photosensitizer to the excited triplet state (T_1_). In this state, phototoxic killing reactions take place by two primary pathways. Type I reactions produce ROS from the reaction of oxygen with radicals formed from the electron transfer reaction of excited photosensitizers and biomolecules that go on to damage cellular structures and organelles. Type II reactions occur when there is a transfer of energy from active photosensitizers to molecular oxygen (^3^O_2_), forming singlet oxygen [19].

There have been several generations of photosensitizers [20,21,22,23]. The first-generation were naturally occurring porphyrins developed in the 1970s. Second-generation photosensitizers were later developed to correct for previous disadvantages, such as prolonged phototoxicity, slow clearance, and dark toxicity. Additional improvements included a red shift to the absorption band to far red (650–690 nm). The increase in wavelength allowed for the treatment of deeper sites in the body. Examples of these include chlorins, benzoporphyrins, and phthalocyanines. Most relevant to this review is the third generation of photosensitizers, which are bioconjugated organic (chemical) photosensitizers with a targeting peptide or antibody (Figure 1; Table 1), thus allowing for increased specificity of phototoxic effects on targeted pathological tissue [24]. More importantly, we want to emphasize the importance of fluorescent protein photosensitizers [25,26,27,28,29,30,31] as a new category of photosensitizer, and their potential use in making novel, targeted fluorescent fusion protein photosensitizers for both PDD and PDT of cancer (Figure 1). It is worth noting that near infrared (NIR) dyes are also being developed for cancer imaging and treatment [32]. For those NIR dyes that possess PDD and/or PDT activity, e.g., IR700 and IRDye800 CW, we also summarized them briefly in this review.

This ability to target pathological tissue without risk of host tissue destruction has made PDT and PDD increasingly appealing in the field of oncology. While current PDD for cancer diagnosis still mainly utilizes non-targeted photosensitizers, there are one well-established and one new potential bioconjugation strategies in PDT (Figure 1). The well-established bioconjugation strategy involves the chemical conjugation of a targeting molecule, which can be a peptide, a full length or partial sequence of a ligand (for a membrane bound receptor), or a full length antibody or antibody fragments (such as single chain variable fragment, scFv), to chemical photosensitizers with direct or indirect linkage. Due to the discovery of fluorescent protein photosensitizers, including KillerRed [26,27], miniSOG (mini singlet oxygen generator) [25,28], SOPP (singlet oxygen photosensitizing protein) [29], and FAP (fluorogen-activating protein) [30,31], here, we propose a potentially new bioconjugation strategy that is the use of recombinant DNA technique to produce fusion proteins that contain a targeting domain and a fluorescent protein photosensitizer simultaneously for targeted PDD and PDT (Figure 1). The aim of this review is to provide a summary of several current bioconjugation targets primarily used in PDT and potentially in PDD of cancer along with an overview of protein photosensitizers.

## 2. Bioconjugation Targets in PDT and PDD of Cancer

Before discussing bioconjugation methods, it is imperative to highlight some of the current targets that are being researched in the targeted PDT and PDD of different cancers. An ideal tumor target molecule would be commonly yet selectively expressed on several malignancy-related compartments in the tumor microenvironment (TME) (Figure 1), including, but not limited to, the cancer cells, tumor vascular endothelial cells (VEC), cancer stem cells (CSC), and myeloid-derived suppressor cells (MDSC). Besides the cancer cells, tumor VEC, the inner layer of tumor neovasculature, is also an important compartment (an estimated ratio of tumor VECs to tumor cells ranging from 1:10 to 1:100) [33,34] in the TME. This is because the tumor neovasculature provides the necessary nutrients and oxygen for cancer cell proliferation. In addition, they are the route by which cancer metastasis is possible. CSC is also a small population (about 0.1–4% in our published study [35]) of neoplastic cells within a tumor, is able to self-renew and develop into the heterogeneous lineages of cancer cells in tumors [36]. CSC plays a role in tumor angiogenesis, resistance to multiple therapies [37,38], recurrence [39,40] and metastasis [35,37,41,42]. MDSC are immature myeloid cells, and they can suppress anti-cancer functions of immune cells, including T cells, dendritic cells and NK cells [43]. Growing evidence suggests that MDSC inhibits the host immune response in cancer patients [44], contributes to resistance to immune checkpoint blockade therapy (ICBT) [45,46], and thus eliminating MDSC may have a positive impact on immunotherapies involving T, NK and dendritic cells [43,46,47,48]. Moreover, tumor-associated macrophages, neutrophils and fibroblasts could be considered as additional tumor compartment targets in the TME as they also play critical roles in tumor progression [49,50]. If a common yet selective surface target molecule can be identified among these tumor compartments, a corresponding targeted therapy may be subsequently developed. Such targeted therapy not only can improve clinical outcomes by targeting multiple tumor compartments, but also may prevent local recurrence and metastasis by targeting CSC and tumor VEC, and overcome resistance to current ICBT by targeting MDSC.

Below is a summary and discussion of several current targets in PDT and/or PDD for a broad range of solid cancers. Our hope is to instill greater appreciation for the potential revolution of tumor-targeted PDT and PDD in the future.

### 2.1. Tissue Factor (CD142)

Tissue factor (TF), also known as CD142 [51,52], is the only membrane-bound coagulation factor (factor III), cofactor, and surface receptor for coagulation factor VII (fVII)/activated fVII (fVIIa) [53,54,55]. The latter is a soluble molecule within the coagulation cascade [56]. Under physiological conditions, TF is not expressed on peripheral blood lymphocytes (T, B, NK, and monocytes) [57,58,59], nor on quiescent VEC of normal blood vessels in normal tissues and organs [60,61,62]. TF expression is restricted to the cells that are not in direct contact with the blood, such as pericytes, fibroblasts, and smooth muscle cells, which are localized in the sub-endothelial vessel wall and thus sequestered from circulating coagulation fVII. TF is key to initiate blood coagulation and to maintain homeostasis. When vessel wall integrity is damaged, TF in pericytes and smooth muscle cells can be bound by fVII and leaked from circulation, forming a TF–fVII complex, which will lead to coagulation cascade activation and eventually clot formation [63].

Accumulated evidence from our laboratory and other groups demonstrate that TF is a common yet selective target for cancer cells [64], tumor VECs [65,66] and CSCs [35] (Figure 2 and Figure 3). First, TF is overexpressed on the cancer cells in many types of solid cancers [64,65,67,68,69], acute myeloid and lymphoblastic leukemia (AML and ALL), and sarcoma [64,69], as well as in Hodgkin’s lymphoma [70] and multiple myeloma [71]. As summarized in our previous reviews [64,69], TF expression was detected on the cancer cells in patients with a broad range of solid cancers, including non-triple breast cancer (non-TNBC) (81–100%), primary melanoma (95%) and metastatic melanoma (100%), lung cancer (40–80%), ovarian cancer (84%), pancreatic cancer (53–90%), colorectal cancer (57–100%), hepatocellular carcinoma (63–100%), primary and metastatic prostate cancer (60–78%), and glioma (47–75%). However, it was previously unknown if TF is expressed in TNBC from patients, and if so, to what extent.

Triple-negative breast cancer (TNBC) is an unmet clinical problem worldwide [72,73], accounting for ~20% of breast cancers. Due to lack of targetable markers (ER, PR and HER2), most cases of TNBC are considered incurable with a shorter life expectancy compared to ER, PR or HER2 positive breast cancer subtypes [72,74,75]. To identify a useful target for TNBC, our laboratory investigated TF expression in TNBC by immunohistochemistry (IHC) in tumor tissue and matched normal breast control tissue from 161 cases of TNBC [65]. The findings demonstrated, for the first time to our knowledge, that TF is expressed on the TNBC cells in up to 85% of patients when using whole tumor tissues (*n* = 14) and ~50% when using tissue microarray (TMA) (*n* = 147) (Figure 2) [65]. Cancer cell TF plays important roles in cancer cell growth, invasion, and metastasis [76,77,78,79,80,81]. For instance, patients with TF-positive tumors often had metastases in the liver, lung, and brain [82], whereas patients with TF negative tumors did not. Second, TF is an angiogenic specific receptor in pathological neovasculature [83]. It is selectively expressed on angiogenic VECs in vitro [83], in vivo in the tumor neovasculature of human melanoma [84,85], lung cancer [86], and breast cancer [87,88] xenografts in mice and in patients’ non-TNBC breast tumors [66]. Using vascular endothelial growth factor (VEGF)-induced in vitro angiogenic VEC models, our lab reported that TF is an angiogenic specific receptor and the target for fVII-targeted immunotherapy and PDT [83]. In TNBC, our recent study [65] showed that TF is not only expressed by the TNBC cells in 50–85% of patients with TNBC (Figure 2 and Figure 3a), but also specifically by tumor VECs (Figure 3b), whereas TF is negative in normal adjacent breast tissues and normal vascular endothelial cells (Figure 2c). The role of angiogenic endothelial TF is a modulator of tumor angiogenesis [33,79,89,90]. TF is strongly linked to cancer progression and has been found to promote tumor growth, metastasis, and angiogenesis in most types of cancers [91,92,93]. Moreover, TF is also selectively expressed on angiogenic VECs in the pathological neovasculature of endometriosis [94], wet form of age-related macular degeneration (AMD) [95]. Thirdly, to our knowledge, for the first time, our lab has demonstrated that TF is also expressed on CSCs (positive for CSC marker CD133, CD133+) isolated from tumor cell lines, xenografts from mice, and patients’ tumor tissues of breast (including TNBC) (Figure 3c), lung, and ovarian cancer [35]. 

The specificity and high expression of TF in cancer cells and angiogenic tumor vascular endothelial cells has made TF a dual therapeutic target, researched initially by Hu and Garen for cancer immunotherapy using an active site-mutated fVII-IgG1 Fc immunoconjugate (called ICON) [84,85,96], with various forms of solid cancers under preclinical studies, which led to the treatment of other pathological angiogenesis-dependent human diseases (notably, AMD [95,97] and endometriosis [94,98]) in preclinical animal models, and eventually to human clinical trials in patients with ocular melanoma (NCT02771340) and AMD (NCT01485588 and NCT02358889) [99]. The Hu laboratory further improved the original ICON (with full length fVII) to L-ICONs (for fVII light chain ICON, to reduce its molecular weight and deplete residual procoagulation activity), called L-ICON1 (second generation) [65] and L-ICON3 (third generation; patent pending), for the treatment of TNBC with enhanced efficacy and safety profiles. Later, TF was also validated as a target in other cancer indications in others works investigating antibody–drug conjugates (ADC) against TF [100,101,102] and in our own work of chimeric antigen receptor-modified NK (CAR-NK) cell therapy [103] under preclinical and clinical investigations.

The Hu laboratory was involved with early research into TF targeted PDT [69,83,86,87,88,104]. The first TF-targeting photosensitizer conjugate was generated by conjugating active-site mutated fVII (K341A) protein to the photosensitizer verteporfin for the PDT treatment of breast cancer in a mouse model [88] and choroidal neovascularization in a rat model [105], the latter is a cause of age-related macular degeneration in humans. The K341A mutation in fVII proteolytic domain significantly reduced its procoagulation activity [65,96,106], while its binding activity to TF retained [96]. The first TF-targeting protein that we constructed for targeted PDT was mfVII/Sp protein [87,88,104], which was composed of murine fVII(K341A), an S peptide (Sp) tag with a mutation at D14N and a polyhistidine tag (His tag, for protein purification and detection) (mfVII(K341A)/Sp(D14 N)/His, abbreviated as fVII/Sp). Later, we made a second TF-targeting protein, fVII/NLS (NLS for nuclear localization sequence) [83,86] with a hope to further bring photosensitizer into nuclear following endocytosis of fVII/TF complex [104]. It is composed of mfVII(K341A) followed by two repeats of the wild-type NLS (PKKKRKVG) of SV40 T-antigen and a His tag [83,86]. The fVII proteins were produced by recombinant DNA technology, specifically, by transfecting Chinese hamster ovary cells (CHO-K1, ATCC) with plasmid encoding fVII/Sp and fVII/NLS cDNAs, whereas verteporfin was extracted from liposomal Visudyne (QLT Inc.) via acidification and separation [88]. EDC (*N*′-3-dimethylaminopropyl-*N*-ethylcarbodiimide hydrochloride) was used as the cross-linker for chemically covalent conjugation of carboxyl groups on the photosensitizers (for example, verteporfin and chlorin e6) and amino groups on the targeting proteins (such as fVII/Sp) [88,105]. High TF expressing MDA-MB-231 cells were selectively killed when treated with mfVII-VP and light. In addition, growth of murine breast cancer tumors was effectively inhibited [88]. Three later studies demonstrated fVII proteins could also be conjugated to the photosensitizer Sn(IV) Ce6 (SnCe6) using EDC for the treatment of breast cancer (including chemoresistant and TNBC) [87,104] and lung cancer [86]. 

Using two breast cancer cell lines with high or low TF level and one TF-negative normal cell line, we showed that the specificity and efficacy of fVII-targeted PDT (fVII-tPDT) using fVII-SnCe6 conjugate was TF-dependent [104]. After having shown, to our knowledge for the first time, that TF is expressed by CD133+ CSC in breast, lung and ovarian cancer [35], we showed that fVII-tPDT could effectively eradicate CSC by the induction of apoptosis and necrosis (Figure 4). CD133+ CSCs was significantly more sensitive to fVII-tPDT than non-CSC cancer cells (CD133 negative, CD133-) (*p* < 0.01) (Figure 4a,b) [35]. Furthermore, fVII-tPDT was effective and safe in vivo against both breast and lung cancer xenograft models with no obvious side-effects [86,87,104]. Selective binding and destruction of angiogenic VECs or tumor vasculature using fVII-SnCe6 tPDT was observed in vitro (Figure 5a–f) and in vivo in those preclinical animal studies [83,86,87,88], similarly via induction of apoptosis and necrosis (Figure 5g) [83,86,87]. Experiments were also done to examine treatment efficacy and specificity of angiogenic VECs (Figure 5), in which angiogenic VECs were stimulated by VEGF and highly expressed TF (with peak expression at 4–6 h) [83], whereas unstimulated VECs were negative for TF expression and used as a normal VEC control [83]. The results showed that fVII-targeted PDT was effective and selective in eradicating angiogenic VECs without harming normal VECs (Figure 5d–f) [83]. This study demonstrated the feasibility of fVII-conjugated photosensitizer as a method of targeting angiogenic VECs for PDT, supporting TF as an angiogenic-specific target in the PDT treatment of cancer [83] and AMD [105], as well as in ICON, L-ICON, and CAR-NK immunotherapy of pathological angiogenesis-dependent diseases, notably cancer [65,84,85,96,103,107], AMD [95,97], and endometriosis [94,98]. 

### 2.2. Human Epidermal Growth Factor Receptor 2

Human epidermal growth factor receptor 2 (HER2) is a receptor tyrosine kinase (RTK) oncogene found in 15–20% of invasive breast cancers and is also associated with gastroesophageal cancers. The HER2 pathway activates multiple signaling cascades that affect cell proliferation, adhesion, migration, apoptosis, and angiogenesis [108]. Early studies demonstrated HER2 to be a poor prognostic indicator for cancer, but the prognosis of HER2-positive cancer has improved since the development of trastuzumab (Herceptin^®^) [109,110]. Trastuzumab is a monoclonal antibody against HER2 often combined with chemotherapy to treat HER2-positive breast cancer. This antibody works by blocking the activation of HER2 on cancer cells, thus inhibiting cell proliferation and cancer progression [111]. While HER2 is expressed in normal adult tissue, the amount is very minimal when compared to the amplification found in breast and gastric cancers and has not affected the efficacy of targeted therapies [112,113,114].

A recent study conjugated the manufactured photosensitizer IRDye700DX (IR700) with trastuzumab to examine PDT of HER2-positive gastric cancer. This treatment strategy was also used in combination with the chemotherapeutic agent 5-fluorouracil (5-FU), as previous studies using PDT alone of gastric cancer demonstrated recurrence. Trastuzumab was incubated with IR700-*N*-hydroxysuccinimide (NHS) in Na_2_HOP_4_ and purified. The resulting IR700-trastuzumab conjugation (Tra-IR700) was used in cell culture and xenograft mouse models. In vitro studies were conducted using high HER2 expressing NCI-N87 human gastric cancer cells and resulted in significant cancer cell destruction with the combined Tra-IR700 and 5-FU therapy. To examine in vivo efficacy, NCI-N87 xenograft tumors were induced in mice. Fluorescence confirmed Tra-IR700 specificity to the HER2 expressing tumor. The intravenous injection of Tra-IR700 was done after the tumors reached 15 mm^3^. In groups with both Tra-IR700 and 5-FU treatment, tumor growth was greatly reduced without recurrence [115].

Another study with trastuzumab examined the use of porphyrin modified trastuzumab in PDT of HER2 gastric cancer. One equivalent of trastuzumab was combined with 10, 20, 30, or 40 equivalents of NHS DMSO cross-linker stock of cationized porphyrin to produce Trast:Porph. The conjugation reaction was conducted in PBS with 3% DMSO and purified. In vitro characterization of PDT using Trast:Porph was done on NCI-N87 cells. The bioconjugation of Trast:Porph resulted in a phototoxic effect specific to HER2-positive cancer cells. In vivo experiments were done on a nude mouse model xenograft with NCI-N87 cells positive for HER2. Greater tumor growth inhibition was found when using Trast:Porph compared to trastuzumab alone [116].

Techniques without trastuzumab have also been effective. Li and colleagues bioconjugated anti-HER2 affibodies with the photosensitizer pyropehophorbide-a (Pyro). HER2-positive NCI-N87 human gastric cancer cells and BT-474 human breast ductal carcinoma cells were used for in vitro experiments. The affibody specific to Z_HER2:2891_ (Z_HER2_) was purified and reduced using TCEP. The photosensitizer Pyro was modified using a peptide polyethylene glycol (PEG) linker to produce Pyro-Linker. This compound was dissolved in DMSO and added to reduced Z_HER2_ in PBS, stirred overnight, and purified. In vitro experiments showed Pyro-Linker-ZHER2 had selectivity towards cancer cell lines with higher HER2 expression. Cancer cells were successfully killed with specificity using PDT. Fluorescence showed in vivo distribution and tumor accumulation of Pyro-Linker-Z_HER2_, further demonstrating its targeting capabilities. High HER2 expression NCI-N87 mouse tumor model was used to examine in vivo PDT activity of Pyro-Linker-Z_HER2_. Once the tumor reached 100 mm^3^, 20 nmol of Pyro-Linker-Z_HER2_ was injected into the mice intravenously along with Pyro and Pyro-Linker-OH control. After about two weeks, tumor disappeared completely, and no recurrence was detected after 40 days of continuous observation [117].

Photosensitizers loaded with anti-HER2 antibody have also been investigated. A previous study demonstrated the use of anti-HER2 indocyanine green (ICG)-doxorubicin (DOX)-loaded polyethyleneimine-coated perfluorocarbon double nanoemulsions (HIDPPDNEs) to examine duel therapy with PDT and chemotherapy. HIDPPDNEs were developed by first generating ICG-DOX-loaded perfluorocarbon double nanoemulsions (IDPPDNEs) via a modified emulsification approach, mixing with PEI, and finally incubating the IDPPDNEs with anti-HER2-mAb. MDA-MB-453 HER2-positive human breast cancer cells were used for in vitro studies. Cytotoxicity was found to be highest with HIDPPDNEs when compared to free ICG and near-infrared (NIR) irradiation alone [118].

Possible use of bioconjugates for guidance of surgical tumor resection and imaging are also under investigation. HER2 targeting antibody antigen-binding fragment (Fab) dual conjugates have been generated for use in live NIR fluorescent imaging and PDT. High quality trastuzumab IgG was digested and reduced, and re-bridged using the cross-linking agent *N*-propargyl3,4-dibromomaleimide to create trastuzumab Fab alkyne. Bio-compatible click chemistry was used to link IRDye800 CW to the alkyne functional group. Only cell lines expressing HER2 demonstrated fluorescence, providing evidence that this technique may be modified in vivo to guide the removal of tumors. The bioconjugate was also linked to the photosensitizer chlorin e6 (Ce6), which demonstrated specific phototoxicity to HER2-positive cells when activated [119]. An additional avenue of HER2-positive cancer imaging is the use of nanoconjugates, which involves linking bioconjugates to nanoparticles (NPs) for delivery into cancer cells. A previous study has demonstrated the specific detection of only HER2-positive cells in vitro using the HER2-positive SKBR3 human breast cancer cell line [120]. This opens avenues for further research into PDD of HER2-positive tumors.

### 2.3. Estrogen Receptor

Another guiding marker for breast cancer treatment is the estrogen receptor (ER) [121]. Similarly to HER2, the ER plays role in the regulation of DNA replication, apoptosis, angiogenesis, and more [122]. Around 75% of all breast cancers are ER-positive, making it a widely used therapeutic target and a crucial factor for subtyping breast cancers [123]. The ER is expressed heterogeneously in normal breast tissue and this depends on the menstrual cycle [124]. However, the overexpression in tumors allows for targeted therapy. The most commonly used agent for the treatment of ER-positive breast cancer is tamoxifen, a selective estrogen receptor modulator, or SERM. Tamoxifen is a prodrug that, when metabolized, forms active metabolites that competitively bind ERs, inhibiting ER-positive cancer cell growth [125]. 

Researchers have conjugated a derivative of tamoxifen with the photosensitizer zinc (II) phthalocyanine to target ER-positive breast cancer cells. These molecules were linked together using a triethylene glycol chain, which through its flexibility increased the biocompatibility and amphiphilicity of the bioconjugate without compromising targeting capabilities. The bioconjugate showed ER mediated targeting to ER-positive human invasive breast ductal carcinoma MCF-7 cells, as the presence of 17β-estradiol downregulated cell uptake in a dose dependent manner. In vitro phototoxicity was observed when compared to controls and was similarly inhibited by the presence of 17β-estradiol in a dose-dependent manner. Highly specific affinity to MCF-7 tumors in Balb/c nude mice was demonstrated [126]. 

Additional research on tamoxifen linked to a photosensitizer has been done. One study developed a tamoxifen modified Ruthenium (II) polypyridyl complex using “click” conjugation. Human breast cancer cell lines MCF-7 and MDA-MB-231, which is of triple negative breast cancer origin, were used. When compared to control, specific phototoxicity was observed against high expressing ER-positive cells but not against MDA-MB-231 cells [127]. Another study used a polymer-drug bioconjugate. Researchers used an oligo-ethyleneglycol linker to bioconjugate tamoxifen to polythiophene. This compound demonstrated selective growth inhibition of MCF-7 cancer cells through ROS formation and subsequent inactivation of ER [128]. A previous and similar study used a tamoxifen-pyropheophorbide bioconjugate to selectively kill MCF-7 cells [129]. Earlier work from this lab showed promise in utilizing estrogen hormone-photosensitizer bioconjugates rather than tamoxifen [130], an additional potential avenue of therapy that has not been investigated further in research at the time of this review article due to the rise in tamoxifen use.

### 2.4. Integrins

Integrins are cell adhesion mediating proteins. They play a role in connecting cells to the extracellular matrix (ECM) and to other cells, but also function in cytoskeleton organization, intracellular signaling, cell proliferation, and migration [131]. Integrin expression has been found to be associated with tumor invasion and cancer progression across different types of cancers, making it a heavily investigated target for cancer therapy [132,133,134].

One of the most widely used targeting peptides for potential integrin targeted cancer therapies is arginine-glycine-aspartic acid (RGD), a small peptide integrin antagonist. A group generated an integrin-targeting silicon (IV) phthalocyanine-cyclic RGD (cRGD) bioconjugate. Silicon (IV) phthalocyanine dichloride was modified and coupled with cRGD via a copper-catalyzed “click” reaction with copper iodide and *N*,*N*-diisopropylethylamine in *N*,*N*-dimethylformamide. Experiments were conducted using HT-29, a high integrin expressing human colon carcinoma cell line, and these demonstrated targeted phototoxicity. In vivo studies using mice with an H22 tumor, which is a high integrin expressing mouse hepatic carcinoma cell line, showed specific localization of the bioconjugate to the tumor under fluorescence and resulted in tumor growth inhibition [135]. Additional research has used a [N,C,N-Pt(II)] complex bearing a N^C^N 1,3-di(2-pyridyl)-benzene linked to a cRGD-tyrosine-lysine peptide to target rat bladder cancer cells [136], EtNBS (a Nile blue derivative) conjugated to cRGD and a PEG chain which showed specific phototoxicity in OVCAR-5 human ovarian cancer cells in vitro [137], and a zinc(II) phthalocyanine substituted with three 2,4-dinitrobenzenesulfonate groups and a cRGD-phenylalanine-lysine moiety that exhibited target phototoxicity to a high integrin expressing cell line MDA-MB-231 cells [138].

Integrin has also been targeted for theranostics of TNBC. One study has used cRGD peptide-decorated conjugated polymer (CP) NPs with poly[2-methoxy-5-(2-ethyl-hexyloxy)-1,4-phenylenevinylene] (MEH-PPV) as the photosensitizer. This bioconjugate was able to specifically identify MDA-MB-231 tumors in mice models in vivo, allowing for PDD of TNBC. In addition, the bioconjugate was able to be activated and destroy the MDA-MB-231 tumor and inhibit growth [139]. This bioconjugate thus has potential to be used for both PDT and PDD in the future and could closely guide tumor removal.

### 2.5. Folate Receptor

Folates play a role in many functions throughout the body. In relation to cancer progression, they play a role in cell proliferation and growth. The folate receptor (FR) was among one of the first targets for cancer therapy, in particularly FR isoform α (FRα). This isoform of FR is the most efficient in folate uptake and has been found to be highly expressed in numerous cancerous, including gastric, lung, brain, endometrial, ovarian, and others [140,141,142]. The expression of the folate receptor is seen in normal human tissues, including kidneys [143], lungs [143], as well as the ovary and liver [144]. However, folate–drug conjugates were found to not accumulate in the lungs [143]. Chemotherapy, metal conjugates, and potentially monoclonal antibodies are some of the targeted therapeutic techniques being developed [145].

One study examined the specificity in a FRα targeted-photosensitizer bioconjugate on a rat NuTu-19 ovarian adenocarcinoma in a rat peritoneal cavity model. Researchers used 5-(4-Carboxyphenyl)-10,15,20-triphenylporphyrin (Porph) and {*N*-{2-[2-(2-aminoethoxy)ethoxy]ethyl} [65,69,71] folic acid}-4-carboxyphenylporphyrin (Porph-s-FA) [144]. Four hours after intraperitoneal administering free PS (Porph) and folic acid (FA)-targeted PS (Porph-s-FA), they sacrificed the animals and examined the correlation of FRα expression (by IHC) and cytoplasmic red fluorescence (as an indication of PS under confocal microscopy) and bio-distribution of PS [144]. The results showed a correlation of FRα expression and fluorescence observation in tumor tissues as well as in several normal tissues (ovary and liver) with a tumor to normal tissue ratio of 9.6 (by assaying porphyrins in nanomoles per gram of protein). Another limitation of the study was that PDT was not tested.

Targeting folate receptor 1 (FOLR1) is also of interest. One study developed porphyrin-lipid NPs (porphysomes) targeted to FOLR1, another FR overexpressed in multiple forms of cancer. Non-targeted porphysomes were first produced by using a lipid film for regular porphysomes consisting of porphyrin-lipid (pyropheophorbide-lipid), cholesterol, and distearoyl-sn-glycero-3-phosphoethanolamine-N-methoxy (PEG) (PEG2000-DSPE). Folate-porphysomes (FPs) were then generated by adding 1,2-distearoyl-sn-glycero-3-phosphoethanolamine-*N*-folate (PEG) to PEG2000-DSPE. This bioconjugate demonstrated in vitro FP-PDT against human lung cancer cells lines A549, H647, H460, and SBC5 with specificity. In vivo studies of FP mediated PDT against A549 tumors in orthotopic lung mice models showed tumor growth inhibition for three weeks following treatment [146].

A similar study also utilized NP delivery in the treatment of mesothelioma by targeting FOLR1. This group combined FP with an epidermal growth factor inhibitor EGFR-tyrosine kinase inhibitor (EGFR-TK1) to inhibit EGFR-associated survival pathways, which has been shown to increase the effects of PDT. FPs were generated using a similar protocol as described above. FOLR1 expressing human mesothelioma cells lines were used for both in vitro and in vivo studies, both demonstrating FOLR1 targeted cell destruction and tumor growth inhibition through FP mediated PDT. Pre-treatment with EGFR-TK1 led to greater phototoxicity, providing evidence for combined treatment efficacy when using PDT [147].

Other studies have focused on modifying folic acid (FA) itself to target folate receptors. Bovine serum albumin (BSA) functionalized chalcopyrite CuFeS_2_ NPs have been bioconjugate with FA and Ce6, creating Ce6:CuFeS_2_@BSA-FA. BSA is able to encapsulate CuFeS_2_ NPs to increase its water solubility with functional groups exposed for FA conjugation. In vitro studies using HeLa cells demonstrated FR targeted PDT. In vitro and in vivo biocompatibility were also examined, and showed minimal cytotoxicity in the absence of light, demonstrating promise for PDD [148]. A different study modified a polydopamine (PDA)-based carrier with FA and loaded it with a cationic phthalocyanine-type photosensitizer (Pc), creating PDA-FA-Pc NPs. This bioconjugate showed remarkable targeted PDT to HeLa and MCF-7 cell lines. In vivo experiments using xenograft mice models with HeLa and MCF-7 tumors also showed specific fluorescence to the tumor, inhibiting growth [149]. Another group conjugated FA with poly(lactide-co-glycolide) polymeric NPs incorporating a photosensitizer, verteporfin, and activated the particles using X-ray radiation. This study demonstrated a possible solution to the low penetration of NIR typically used in PDT with X-ray radiations excellent penetration. FA targeted PDT was tested on HCT116 human colon cancer cells, showing enhanced uptake and phototoxicity [150].

### 2.6. Epidermal Growth Factor Receptor

EGFR (also known as ERBB1 or HER1) is an RTK that plays a crucial role in the homeostasis and growth of epithelial cells [151]. Its role in tumor progression in multiple carcinomas is well established, as the RTK has been shown to promote tumor proliferation, angiogenesis, metastasis, and more. Multiple types of cancer have been found to exhibit over expression of EGFR, making it a candidate for targeted therapy. Current therapeutic interventions specific to EGFR include treatment of lung, pancreatic, and colorectal cancers, among others [152,153,154,155].

The use of EGFR-targeted PDT is currently under investigation. One group used NB 7D12 and NB 7D12-9G8 nanobodies that target EGFRs conjugated with the photosensitizer IR700. In vitro PDT was found to be specific to high EGFR expressing squamous cell carcinoma of the tongue 19-luc2-cGFP cells. In vivo experiments demonstrated tumor necrosis when treated with the bioconjugate [156]. Researchers have also developed an EGFR targeting nanobody for PDT of human head and neck squamous cell carcinoma. Nanobodies NB 7D12, NB 7D12-9G8, and the mAb cetuximab were each conjugated to IR700. UM-SCC-14C cells were used to establish phototoxic effects on high EGFR expressing cells with each conjugate. Patient derived tumor organoids were also treated with the conjugates and were killed with efficiency depending on the amount of EGFR expression [157]. PDT targeted to EGFR for gastric cancer has also been explored. EGF-conjugated chitosan NPs were crosslinked with tripolyphosphate (TPP) and curcumin, a photosensitizer, to generate curcumin-encapsulated and EGF-conjugated chitosan/TPP NPs to treat MKN45 human gastric cancer cells. In vitro PDT was accomplished with specificity [158]. Another team used the SNAP-tag to conjugate single chain variable fragment (ScFv)-425 Fab to the photosensitizer Ce6 to target EGFR. Carboxyl groups of Ce6 were modified with benzylguanine linker to allow for binding to the SNAP-tag. In vitro phototoxicity of scFv-425-SNAP-Ce6 was evaluated using the EGFR-positive cell lines A431, MDA-MB-231, and SiHa (cervical carcinoma), showing targeted PDT [159].

Additional research is focusing on theranostics of triple-negative breast cancer. EGFR antibodies have been conjugated to IR700 phthalocyanine using the SNAP-tag. This construct has demonstrated the ability to image high EGFR expressing cancer cell lines and human tissue samples. It also provided a method of treatment that can easily be initiated, thus combining PDT and PDD [160].

### 2.7. P-glycoprotein

Multidrug resistance (MDR) is a major challenge when treating carcinomas. P-glycoprotein (Pgp) is an efflux transporter found in normal cells, but in cancer cells it contributes to MDR by transporting drugs out of the cell, thus preventing drug action and treatment. Pgp has been found to influence MDR in a variety of cancers, including liver, lung, skin, and more [161,162,163]. Inhibiting Pgp is of particular interest in oncology in order to overcome MDR [164]. Pgp targeted PDT may be able to bypass drug resistance, as it allows for direct destruction of the tumor promoting transporter [165].

One such study demonstrated the use of Pgp targeted PDT to eliminate chemoresistant cancer cell lines. Researchers conjugated the monoclonal antibody 15D3, an anti-Pgp antibody (Pab), to IR700. Purified Pab was incubated with IR700-NHS to produce the bioconjugate Pab-IR700. In vivo xenograft mouse models baring chemoresistant tumors were successfully treated [166]. A similar follow-up study was done using Fab-photosensitizer conjugates against Pgp, with similar therapeutic efficacy [167]. The same lab used Pab-IR700 to prime the tumor microenvironment for nanomedicine delivery. Multiple cell lines were used in this study, including cells that were Pgp positive/negative and/or chemo-sensitive/resistant. Cell lines positive for Pgp were destroyed by Pab-IR700. Combined chemotherapy with PDT was also examined in vitro, showing improved treatment efficacy in chemoresistant cell lines when compared to chemotherapy alone [168].

Another method of targeting Pgp with PDT is the use of nanotubes and NPs. Researchers engineered multiwalled carbon nanotubes (MWCNTs) surrounded with a dense coating of phospholipid-poly(ethylene glycol). Pab was then conjugated with *N*-Succinimidyl S-acetylthioacetate and deacetylated to generate thiolated Pab. This was then reacted with MWCNTs-PEG-maleimide to form the final bioconjugate. High Pgp expressing 3T3-MDR1 cells (mouse fibroblasts cell line with transfected Pgp cDNA) were selectively killed by the conjugate upon light irradiation. An additional experiment demonstrated targeted phototoxic destruction of NCI/ADR-RES tumor spheroids, an adriamycin-resistant ovarian cancer cell line with high Pgp expression [169]. An overview of each target with its respective bioconjugation method is briefly summarized in Table 1.

## 3. Bioconjugation Strategies

### 3.1. Chemical Linkage Methods

Several major routes of bioconjugation using linkage pathways have been described, with the most relevant to PDT being cysteine coupling, lysine coupling, and “click” chemistry [24]. These have been described at length in other reviews [170,171], and so will be covered briefly here. Reduced disulfide bridges can yield free thiols for cysteine route conjugation. Maleimides are an example that uses this cysteine route by reacting with thiols on reduced targeting IgG [172]. A lysine coupling route utilized in bioconjugation is amide conjugation. Perhaps the most utilized method is the use of photosensitizers activated with NHS esters to bind amines on lysine residues of targeting peptides and antibodies [24]. Click chemistry is most often via the copper-catalyzed azide-alkyne cycloaddition (CuAAC) reaction. This route of conjugation is becoming increasingly used due to the inherent benefits of the 1,2,3-triazole linker. This provides more favorable conditions in vivo, stability, and places the linked components in orientations away from each other to prevent [173]. SNAP-tag conjugation is another promising method of bioconjugation. The SNAP-tag is a self-labeling protein and a modified alkylguanine-DNA alkyltransferase (AGT), a human DNA repair enzyme. This enzyme reacts with *O*(6)-benzylguanine (BG)-modified molecules with specificity through alkyl group transfer to a cysteine. Thus, the technology is a convenient and easy way to tag antibodies for PDT and PDD of cancer [174,175,176]. Another route of conjugation is the use of commercially available antibody-oligonucleotide kits. One such example labeling an amino-modified oligonucleotide with succinimidyl-4-formylbenzamide. The formylbenzamide within the amine reactive NHS ester can then be linked to an antibody modified by succinimidyl-6-hydrazino-nicotinamide, a complementary linker. The conjugation is completed with the addition of aniline and finally purification [177].

### 3.2. Recombinant DNA Technology for Making Fluorescent Fusion Protein Photosensitizers

While chemical conjugates are the most widely used, there are some limitations in the chemical conjugation technique, including poor reproducibility, formation of aggregation and impurity with unconjugated (free) photosensitizers [27,178]. Additionally, these conjugates are unable to be reproduced within the target tissue [179]. Emerging technologies are being investigated to overcome these shortfalls, in particular the use of protein photosensitizers.

In 2006, one of the first genetically encoded photosensitizers KillerRed was identified [26]. GFP analogs in *Escherichia coli* cells transformed with the pWE30 vector were screened for phototoxic effects. Once KillerRed was identified, several mutations to increase its efficacy were induced [26]. KillerRed is of increased interest due to its ease in production over chemical conjugation and the ability of protein photosensitizers to be transfected and expressed in target cells in desired cellular compartments [180]. Several studies into the use of KillerRed for PDT of cancer have been performed. Purified KillerRed protein has been used to kill multiple leukemia cell lines, including K562, NB4, and THP1 [181]. In vivo studies have also been conducted. In 2012, tumor mouse models bearing a HeLa Kyoto cell line expressing KillerRed in fusion with histone H2B were successfully destroyed upon light irradiation. Lentiviral transduction was utilized to induce expression of the photosensitizer within the mitochondria of the tumor cells [182]. Another study in 2015 also used KillerRed in fusion with histone H2B, and expression was within CT26 murine colon carcinoma subcutaneous mouse tumors through lentiviral infection. Selective phototoxicity was also achieved [183]. Vectors encoding KillerRed, pKillerRed-mem, have been delivered to epithelial cells using chitosan (CS) and poly (γ-glutamic acid) (γPGA) complexes. Fluorescence demonstrated expression of the protein photosensitizer in the cell membrane of target cells. Subsequent light irradiation showed phototoxic killing of the KillerRed expressing cells, providing evidence for use of this strategy in the treatment of various pathologic tissues beyond cancer [184]. A variety of different engineered protein photosensitizers exist [185], and their therapeutic efficacy is worth exploring.

Besides those studies of target cell expression of KillerRed, a fusion protein of anti-p185^HER-2-ECD^ 4D5scFv and KillerRed was produced by transforming E. coli SB536 with plasmid pSD-4D5scFv KillerRed [27]. The ovarian carcinoma cell line SKOV-3, which has high expression levels of p185^HER-2-ECD^, were destroyed with specificity upon light irradiation (bright white light for 10 min, ~1 W/cm^2^, equivalent to ~0.2 W/cm^2^ of KillerRed-activating green light) [27]. Compared to KillerRed alone, the excitation and emission spectra of scFv-KillerRed conjugate remained unaltered [27]. However, the in vivo efficacy and safety of fluorescent protein photosensitizer conjugates remains to be evaluated in animal models.

Another genetically encoded protein photosensitizer is green fluorescent flavoprotein miniSOG (mini singlet oxygen generator) [28]. miniSOG is a monomeric peptide with 106 amino acid residues and can generate singlet oxygen upon blue-light illumination [25]. This protein photosensitizer typically kills cells via the type II photoreaction [186].

One study has used miniSOG to target HER2-positive breast cancer cells SK-BR-3 using the 4D5scFv targeting molecule [187]. Researchers genetically engineered a recombinant protein 4D5scFv-miniSOG, which is selectively targeted to the extracellular domain of HER2. In vitro specific phototoxicity was seen against high HER2 expressing SK-BR-3 cells [187].

Another study targeted the same cell line but instead used the DARPin-9_29 (Designed Ankyrin Repeat Proteins) [188], a non-immunoglobulin targeting molecule to HER2. The DARPin-miniSOG bioconjugate was genetically engineered and selectively killed SK-BR-3 cells. A drawback to using this conjugate was a reduction in cytotoxicity, likely due to the more rapid internalization and degradation of DARPin-miniSOG [188].

Researchers have also developed fusion proteins that contain no linker molecule between the targeting peptide and photosensitizer. One group developed a fusion protein containing the urokinase receptor targeting agent and β-carboxy phthalocyanine zinc (ZnPc-COOH) (CPZ) to target urokinase-type plasminogen activator receptor (uPAR), which is overexpressed in many invasive carcinomas. NPs were used for delivery of the fusion protein to H22 mouse hepatocellular carcinoma cell line tumors in mice, resulting in tumor growth inhibition [189].

## 4. Conclusions

PDT and PDD are emerging techniques in the diagnosis and treatment of cancer. Many different bioconjugation techniques of targeting molecules with photosensitizers exist, with the most prominently used being chemical conjugation using linker chemistry. While most published literature that we discussed here focuses on PDT, the potential application for PDD may also be tested if the photosensitizer in the bioconjugates can emit fluorescence upon excitation of a suitable wavelength of light. Fluorescent protein sensitizers and fusion proteins are a promising avenue of bioconjugation and may address some of the current limitations with regard to chemical conjugation. Future research should be focused on improving these strategies and expanding the use of novel, dual-action bioconjugates with the ability to simultaneously diagnose and treat, ideally by targeting multiple tumor compartments, a wide range of solid cancers, and blood-borne malignancies.

## Figures and Tables

**Figure 1 molecules-25-04964-f001:**
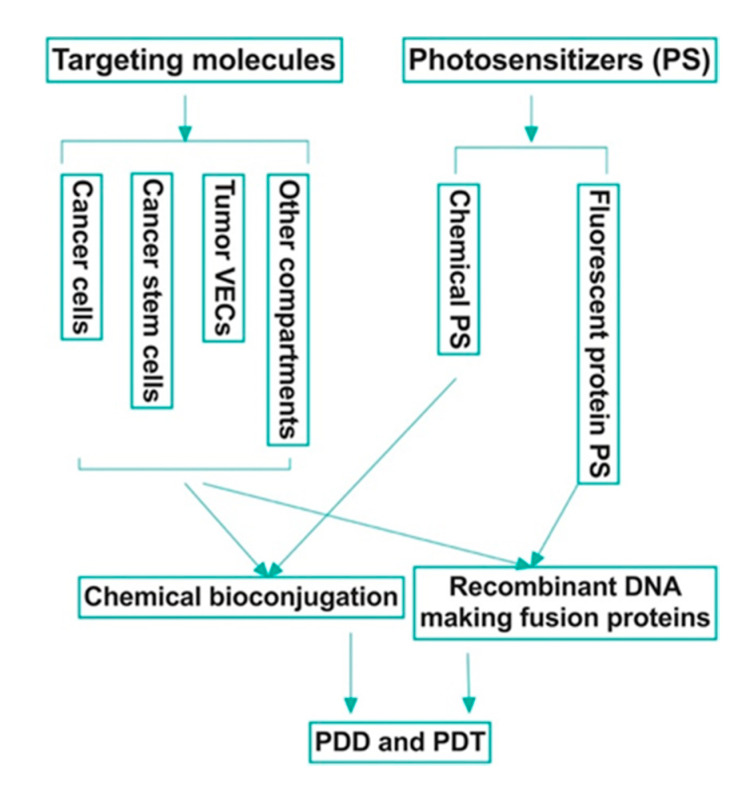
Bioconjugation strategies in photodynamic diagnosis (PDD) and therapy (PDT) of cancer. An ideal target, such as tissue factor (also known as CD142), should be commonly yet selectively expressed by multiple tumor compartments, including but not limited to, the cancer cells, cancer stem cells and tumor vascular endothelial cells, whereas it is negatively, minimally or restrictedly expressed in normal cells.

**Figure 2 molecules-25-04964-f002:**
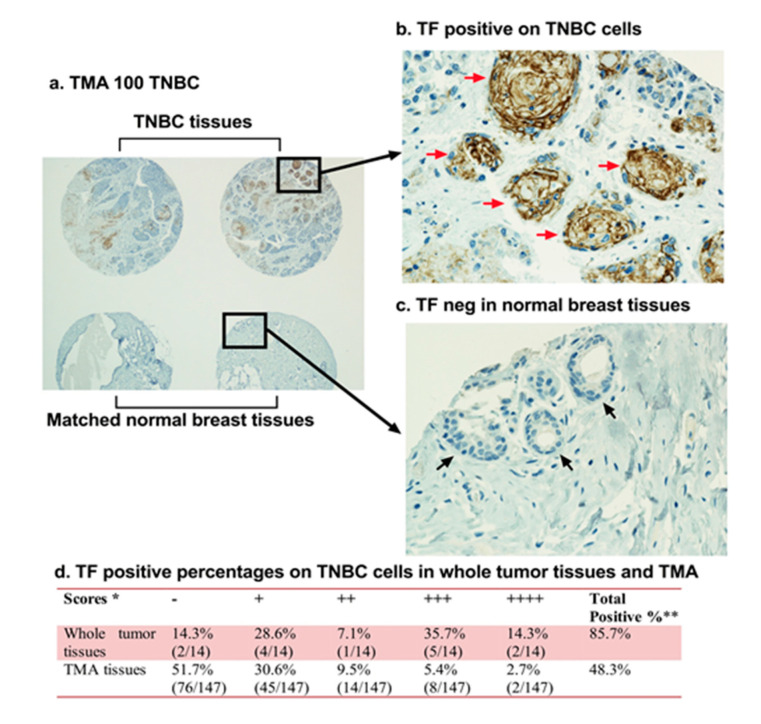
Tissue factor (TF) is a new surface target in 50–85% of TNBC patients. TF expression in 161 patients’ TNBC was examined by immunohistochemical (IHC) staining on Tissue Microarray (TMA) slides with TNBC tissues (*n* = 147) (**a**,**b**,**d**) and matched normal breast tissues (**a**,**c**) and whole tumor tissues (*n* = 14) (**d**). * The Scores for TF expression were graded as follows: negative (−), moderately positive (+), positive (++), strongly positive (+++) and very strongly positive (++++). Positive percentages included all cases graded from moderately positive through very strongly positive. ** Fisher’s exact test was used to test IHC score percentage difference between whole tumor tissues and TMA tissues, there is a significant difference with *P* = 0.0002474. Modified from Hu et al. *Cancer Immunol Res*. 2018 (doi: 10.1158/2326-6066.CIR-17-0343).

**Figure 3 molecules-25-04964-f003:**
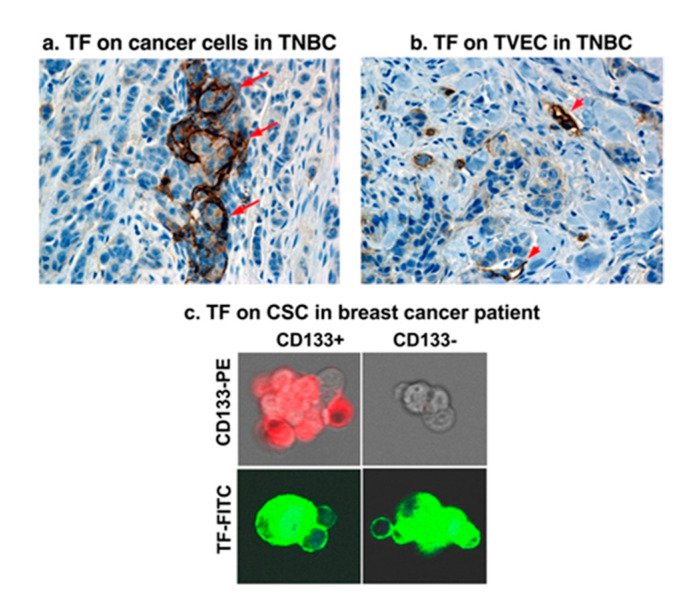
TF is expressed commonly on multiple compartments in the TNBC microenvironment. (**a**) TF on the TNBC cancer cells (TF positive staining in brown, arrows). (**b**) TF on tumor vascular endothelial cells (TVECs) (TF positive staining in brown, arrowheads). (**c**) TF on cancer stem cells (CSCs). Panels a & b: Modified from Hu et al. *Cancer Immunol Res*. 2018 (doi: 10.1158/2326-6066.CIR-17-0343)**;** Panel c: Modified from Hu et al. *Oncotarget* 2017 (doi: 10.18632/oncotarget.13644).

**Figure 4 molecules-25-04964-f004:**
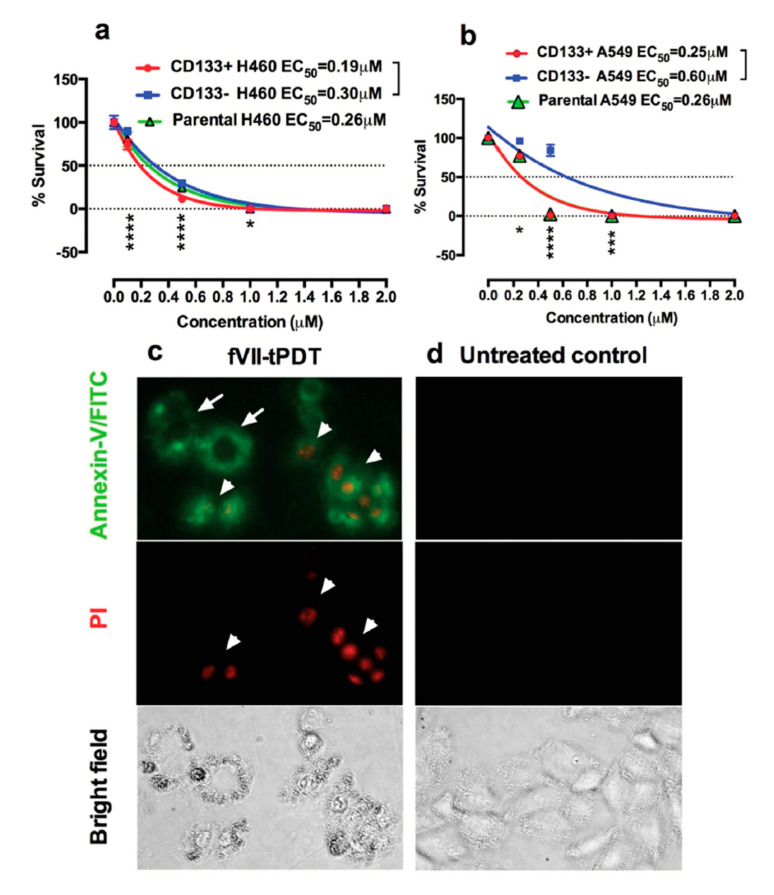
TF-targeted PDT using fVII-SnCe6 chemical conjugate was effective in eradicating CD133+ CSCs and CD133- non-CSC cancer cells via induction of apoptosis and necrosis. a,b: % survival of CD133+ CSCs, CD133- non-CSC and parental cancer cells H460 (**a**) and A549 (**b**) after being treated by fVII-tPDT for 36 J/cm^2^ 635 nm laser light, as determined by clonogenic assay. *: *p* < 0.05; ***: p* < 0.01; ***: *p* < 0.001; ****: *p* < 0.0001. (**c**,**d**): After fVII-tPDT, the CD133+ H460 CSC cells (**c**) were stained with Annexin V-FITC and then stained with propidium iodide (PI). Untreated CD133+ H460 CSCs were the control cells (**d**). The cells were photographed under a fluorescent microscope using green (Annexin/FITC for apoptosis, white arrows), red (PI for necrosis, white arrowheads) and bright field channels. Original magnification: 200×. Modified from Hu et al. *Oncotarget* 2017 (doi: 10.18632/oncotarget.13644).

**Figure 5 molecules-25-04964-f005:**
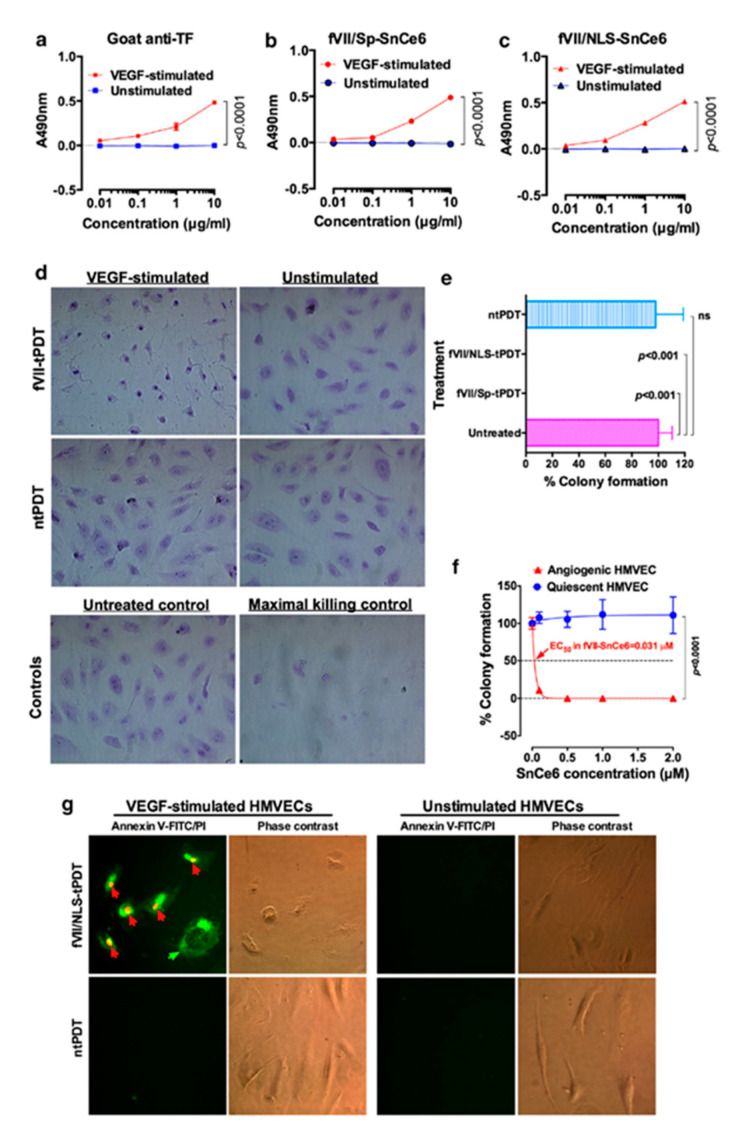
TF-targeted PDT (fVII-tPDT) using chemically conjugated fVII-SnCe6 is selective and effective in eradicating angiogenic VEC via binding to TF and by induction of apoptosis and necrosis. (**a**–**c**). fVII-SnCe6 conjugates retain the binding activity and selectivity to angiogenic VECs (VEGF-stimulated HMVEC: human microvascular endothelial cells), while it has no binding to unstimulated HMVEC. Goat anti-TF was a positive control (**a**), SnCe6 was separately conjugated with mfVII/Sp (**b**) and mfVII/NLS (**c**). (**d**) Representative imaging of crystal violet stained VEGF-stimulated and unstimulated HMVECs right after being treated with fVII-tPDT or ntPDT (2 µM and 635 nm laser light at 36 J/cm^2^). Control HMVECs include an untreated control and a maximal killing control (completely lysed by 1% Triton X-100). (**e**) Complete eradication (no colonies formed) of angiogenic VEC (HMVEC) by fVII-tPDT using fVII/NLS-SnCe6 or fVII/Sp-SnCe6, whereas ntPDT has no therapeutic effect in killing angiogenic VEC. (**f**) The fVII-tPDT is effective and selective in killing angiogenic VEC, whereas it has no side effects on quiescent VEC (635 nm laser light at 36 J/cm^2^). Note that the VEC cells without fVII/NLS-SnCe6 (0.0 µM) also served as the light only control as they were also irradiated with 635 nm laser light (36 J/cm^2^). (**g**) The underlining mechanism of fVII/NLS-tPDT involves rapid induction of apoptosis and necrosis. Annexin V-FITC (green) stains for apoptotic cell membrane (green arrow), while propidium iodide (PI, red) stains for the nuclei of dead cells (red arrows). Modified from Hu et al. *Angiogenesis*. 2017 (doi: 10.1007/s10456-016-9530-9).

**Table 1 molecules-25-04964-t001:** Summary of current targets and chemical bioconjugation strategies in PDT and PDD of cancer.

Target	Targeting Molecule	Photosensitizer	Conjugation Method	Cell Lines/Tumor Xenografts in Mice	Reference
TF (CD142)	Active-site mutated fVII	SnCe6	EDC linker	A549, H460	[86]
TF (CD142)	Active-site mutated fVII	Verteporfin	EDC linker	MDA-MB-231 (human TNBC), VEGF-stimulated HUVEC vs. unstimulated HUVEC, murine breast cancer EMT6 xenografts	[88]
TF (CD142)	Active-site mutated fVII	SnCe6	EDC linker	Chemosensitive (MCF-7) and chemoresistant (MCF-7/MDR) breast cancer	[87]
TF (CD142)	Active-site mutated fVII	SnCe6	EDC linker	MDA-MB-231, VEGF-stimulated HUVEC vs. unstimulated HUVEC, murine breast cancer EMT6 xenografts	[104]
HER2	Trastuzumab	IR700	NHS ester reaction	NCI-N87	[115]
HER2	Trastuzumab	Porphyrin	NHS ester reaction	NCI-N87	[116]
HER2	Anti-HER2 affibody	Pyropheophorbide-a	PEG linker	NCI-N87, BT-474	[117]
HER2	Anti-HER2 antibody	ICG-DOX	PEI linker	MDA-MB-453	[118]
HER2	HER2 Fab	IRDye800 CW	Maleimide conjugation	OE19	[119]
HER2	Trastuzumab	Zinc tetracarboxyphenoxy phthalocyanine	NHS ester reaction	SKBR3	[120]
ER	Tamoxifen	Zinc (II) phthalocyanine	Triethylene glycol chain	MCF-7	[126]
ER	Tamoxifen	Ruthenium (II) polypyridyl	“Click” conjugation	MCF-7	[127]
ER	Tamoxifen	Porphyrin	Oligo-ethyleneglycol linker	MCF-7	[128]
ER	Tamoxifen	Pyropheophorbide	Nucleophilic substitution	MCF-7	[129]
ER	C17-α-alkynylestradiol	Ce6-dimethyl ester	Nucleophilic substitution	MCF-7	[130]
Integrin	RGD	Phthalocyanine	“Click” reaction	HT-29, H22	[135]
Integrin	RGD	Platinum (II) complexes	NHS ester reaction	AY27	[136]
Integrin	RGD	EtNBS	“Click” reaction	OVCAR-5	[137]
Integrin	RGD	Zinc (II) phthalocyanine	“Click” reaction	A549, MDA-MB-231	[138]
Integrin	RGD	MEH-PPV	Maleimide conjugation	MDA-MB-231	[139]
FRα	FA	Porphyrin	NHS ester reaction	NuTu-19	[144]
FOLR1	FA	Porphyrin	PEG linker	A549, H647, H460, SBC5	[146]
FOLR1	FA	Porphyrin	PEG linker	AE17, AE17-sOVA, AK7, AB12, RN5, H28, H226, H2052, H2452	[147]
FR	FA	Ce6	NHS ester reaction	HeLa	[148]
FR	FA	Phthalocyanine	Polydopamine nanomedicine	HeLa, MCF-7	[149]
FR	FA	Verteporfin	NHS ester reaction	HCT116	[150]
EGFR	NB 7D12 and NB 7D12-9G8 nanobodies	IR700	NHS ester reaction	19-luc2-cGFP	[156]
EGFR	NB 7D12 and NB 7D12-9G8 nanobodies, cetuximab	IR700	NHS ester reaction	UM-SCC-14C	[157]
EGFR	EGF-conjugated chitosan nanoparticles	Curcumin	NHS ester reaction	MKN45	[158]
EGFR	scFv-425 Fab	Ce6	SNAP-tag	A431, MDA-MBDA-231, SiHa	[159]
EGFR	Anti-EGFR antibody	IR700	SNAP-tag	Hs758T, MDAMB-231, MDA-MB-453, MDA-MB-468	[160]
Pgp	Pab	IR700	NHS ester reaction	3T3-MDR1, NCI/ADR^Res^, KB-8–5-11	[166]
Pgp	Pgp Fab	IR700	Maleimide conjugation	3T3-MDR1, KB-8-5-11	[167]
Pgp	Pab	IR700	NHS ester reaction	3T3-MDR1, NCI/ADR^Res^, KB-8–5-11	[168]
Pgp	Pab	MWCNT nanomedicine	Maleimide conjugation	3T3-MDR1	[169]

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
