# Peer review of "Current Targets and Bioconjugation Strategies in Photodynamic Diagnosis and Therapy of Cancer"

_molecules, 2020, doi:10.3390/molecules25214964_

Round 1
Reviewer 1 Report
Overall:
This article reviews some current promising target molecules utilized in bioconjugation reactions and strategies for the photodynamic diagnosis and treatment of various cancers, with a special focus on protein photosensitizers.
This paper is novel, relevant and of high significance, since providing an overview of treatment targets and bioconjugation strategies will aid in driving research in this field forward towards more effective, less toxic and non- or minimally invasive treatment and diagnosis options for cancer patients.
The paper is well written, excellent and flawless. Authors have organized this paper systematically with a clear understanding and interpretation of high-quality data.
The overall quality of the article is good with scientific rigor and I think it will be highly cited due to its significance, relevance and need. This study is novel and original and so is of scientific merit. I do commend this paper and its contribution to the journal and personally believe it will be of interest to its readers, thus it should be accepted as is, with only a few minor revisions required.
Minor Suggested Changes:
Line 79 to 80: CSC is a small subpopulation of neoplastic cells within a tumor that theoretically possess the capacity to self-renew and develop into the heterogeneous lineages of cancer cells which are found in tumors [16]. Remove the word comprise
Line 86 to 91: If such a common yet specific surface target molecule can be discovered on these tumor compartments, corresponding targeted therapies could be developed so that these targeted therapies not only can achieve an optimal clinical outcome (by simultaneously targeting several or all these malignancy-related compartments), but also prevent local recurrence and metastasis (by targeting CSCs and tumor VECs), as well as overcome resistance to current therapies, such as immune checkpoint blockade therapy (by targeting suppressor cells such as MDSCs). Reword sentence
Line 420 to 421: most relevant to PDT being cysteine coupling, lysine coupling, and “click” chemistry [14]. These have been described at length in other reviews [145,146], and so will be briefly covered here.
Author Response
Title: Bioconjugation Strategies in Photodynamic Diagnosis and Therapy of Cancer
We would like to thank the reviewer for their insightful comments and believe that by addressing them we have greatly improved the review for the visibility and impact. We are hopeful that the revisions satisfy the concerns raised in the review. The response to the critique can be found below.
Reviewer 1: Comments and Suggestions for Authors
Overall:
This article reviews some current promising target molecules utilized in bioconjugation reactions and strategies for the photodynamic diagnosis and treatment of various cancers, with a special focus on protein photosensitizers.
This paper is novel, relevant and of high significance, since providing an overview of treatment targets and bioconjugation strategies will aid in driving research in this field forward towards more effective, less toxic and non- or minimally invasive treatment and diagnosis options for cancer patients.
The paper is well written, excellent and flawless. Authors have organized this paper systematically with a clear understanding and interpretation of high-quality data.
The overall quality of the article is good with scientific rigor and I think it will be highly cited due to its significance, relevance and need. This study is novel and original and so is of scientific merit. I do commend this paper and its contribution to the journal and personally believe it will be of interest to its readers, thus it should be accepted as is, with only a few minor revisions required.
Minor Suggested Changes:
Comment 1. Line 79 to 80: CSC is a small subpopulation of neoplastic cells within a tumor that theoretically possess the capacity to self-renew and develop into the heterogeneous lineages of cancer cells which are found in tumors [16]. Remove the word comprise
Response: We appreciate the comment and have removed the word “comprise” and have rewritten the sentence.
“CSC is also a small population (about 0.1-4% in our published study [35]) of neoplastic cells within a tumor, is able to self-renew and develop into the heterogeneous lineages of cancer cells in tumors [36].”
Comment 2. Line 86 to 91: If such a common yet specific surface target molecule can be discovered on these tumor compartments, corresponding targeted therapies could be developed so that these targeted therapies not only can achieve an optimal clinical outcome (by simultaneously targeting several or all these malignancy-related compartments), but also prevent local recurrence and metastasis (by targeting CSCs and tumor VECs), as well as overcome resistance to current therapies, such as immune checkpoint blockade therapy (by targeting suppressor cells such as MDSCs). Reword sentence
Response: Thank you for your comments. We have reworded the sentence as seen below as well as in the revised manuscript.
“If a common yet selective surface target molecule can be identified among these tumor compartments, corresponding targeted therapy may be subsequently developed. Such targeted therapy not only can improve clinical outcomes by targeting multiple tumor compartments, but also may prevent local recurrence and metastasis by targeting CSC and tumor VEC, and overcome resistance to current ICBT by targeting MDSC.”
Comment 3. Line 420 to 421: most relevant to PDT being cysteine coupling, lysine coupling, and “click” chemistry [14]. These have been described at length in other reviews [145,146], and so will be briefly covered here.
Response: We really appreciate your suggestions. We have adapted them in the revised manuscript, as seen below.
“with the most relevant to PDT being cysteine coupling, lysine coupling, and “click” chemistry [24]. These have been described at length in other reviews [170,171], and so will be briefly covered here.”
Reviewer 2 Report
The review paper aims to provide a summary of the current bioconjugation strategies used in photodynamic diagnosis and therapy of cancer. The paper highlighted seven protein molecules used for selective targeting and cytotoxicity using PDT. Lastly, the potential of genetically encoded fluorescent proteins as photosensitizers was discussed.
- A few schematic diagrams and histograms could be very helpful for easy visualization and elucidation of ideas.
- Lines 46-49 describes the primary pathways of PDT’s action. More precisely, type I reactions do yield reactive superoxide species (ROS) from the reaction of oxygen with radicals formed from the electron transfer reaction of excited PSs and biomolecules. Type II reactions categorically produce singlet oxygen through energy transfer from the PS to molecular oxygen.
- How did the authors decide on which proteins overexpressed in cancer cells to feature in the paper?
- Lines 105-108 are repeating ideas presented in the previous statements. They can be removed for clarity.
- For consistency, lines 164-165, 200, 211, 223 may be removed as they are purification methods which are not stated anywhere else in the paper.
- Minor changes: In lines 111-112, the phrase “In cancer” may be removed or the sentence can be paraphrased. “Accumulating” should be “Accumulated” and the word “oncotarget” does not exist but as a name of a journal. In line 170, the phrase “in mice” is redundant. In line 228, tumor volume reached 100 “mm3” not “m3”. In line 233, the word “duel” should be “dual”. In line 337, “orthotropic” should be “orthotopic”.
- Please discuss more on this topic found in line 395, “PDT is a method of bypassing Pgp MDR currently being researched to improve the efficacy of cancer therapy”.
Author Response
Reviewer 2
Comments and Suggestions for Authors
The review paper aims to provide a summary of the current bioconjugation strategies used in photodynamic diagnosis and therapy of cancer. The paper highlighted seven protein molecules used for selective targeting and cytotoxicity using PDT. Lastly, the potential of genetically encoded fluorescent proteins as photosensitizers was discussed.
Comment 1. A few schematic diagrams and histograms could be very helpful for easy visualization and elucidation of ideas.
Response: We appreciate your great suggestions and comments. We have included a few new figures (Figures 1-5) for visualization and elucidation of ideas (Figure 1) and for highlighting the specificity and effectiveness for tissue factor as a useful target that is commonly yet selectively expressed on cancer cells, cancer stem cells and angiogenic tumor vascular endothelial cells in targeted PDT (Figure 2-5).
While one point we want to promote identification of target molecules on multiple tumor compartments for the development of targeted PDD and PDT, here we also attempt to emphasize the importance of recently discovered fluorescent protein photosensitizers in potential use as dual agents for both PDD and PDT. Therefore, we have rewritten the third and fourth paragraphs (page 2) under Introduction.
“More importantly, we want to emphasize the importance of fluorescent protein photosensitizers [25-31], as a new category of photosensitizer, and their potential use in making novel, targeted fluorescent fusion protein photosensitizers for both PDD and PDT of cancer (Figure 1). It is worth noting that near infrared (NIR) dyes are also being developed for cancer imaging and treatment [32]. For those NIR dyes that possess PDD and/or PDT activity, for example, IR700 and IRDye800 CW, we also summarized them briefly in this review (Table 1).
This ability to target pathological tissue without risk of host tissue destruction has made PDT and PDD increasingly appealing in the field of oncology. While current PDD for cancer diagnosis still mainly utilizes non-targeted photosensitizers, there are one well-established and one new potential bioconjugation strategies in PDT (Figure 1). The well-established bioconjugation strategy is chemical conjugation of a targeting molecule, which can be a peptide, a full length or partial sequence of a ligand (for a membrane bound receptor), a full length antibody or antibody fragments (such as single chain variable fragment, scFv), to chemical photosensitizers with direct or indirect linkage. Due to the discovery of fluorescent protein photosensitizers, including KillerRed [26,27], miniSOG (mini singlet oxygen generator) [25,28], SOPP (singlet oxygen photosensitizing protein) [29] and FAP (fluorogen-activating protein) [30,31], here we propose a potentially new bioconjugation strategy that is the use of recombinant DNA technique to produce fusion proteins that contain a targeting domain and a fluorescent protein photosensitizer simultaneously for targeted PDD and PDT (Figure 1). The aim of this review is to provide a summary of several current bioconjugation targets primarily used in PDT and potentially in PDD of cancer along with an overview of protein photosensitizers.”
Figure 1. Bioconjugation strategies in photodynamic diagnosis (PDD) and therapy (PDT) of cancer. An ideal target, such as tissue factor (also known as CD142), should be commonly yet selectively expressed by multiple tumor compartments, including but not limited to, the cancer cells, cancer stem cells and tumor vascular endothelial cells, whereas it is negatively, minimally or restrictedly expressed in normal cells.
Figure 2. Tissue factor (TF) is a new surface target in 50-85% of TNBC patients. TF expression in 161 patients’ TNBC was examined by immunohistochemical (IHC) staining on Tissue Microarray (TMA) slides with TNBC tissues (n=147) (a, b, d) and matched normal breast tissues (a, c) and whole tumor tissues (n=14) (d). Modified from Hu et al. Cancer Immunol Res. 2018 (doi: 10.1158/2326-6066.CIR-17-0343)
Figure 3. TF is expressed commonly on multiple compartments in the TNBC microenvironment. a. TF on the TNBC cancer cells. b. TF on tumor vascular endothelial cells (TVECs). c. TF on cancer stem cells (CSCs). Panels a & b: Modified from Hu et al. Cancer Immunol Res. 2018 (doi: 10.1158/2326-6066.CIR-17-0343); Panel c: Modified from Hu et al. Oncotarget 2017 (doi: 10.18632/oncotarget.13644)
Figure 4: TF-targeted PDT using fVII-SnCe6 chemical conjugate was effective in eradicating CD133+ CSCs and CD133- non-CSC cancer cells via induction of apoptosis and necrosis. a-b. % survival of CD133+ CSCs, CD133- non-CSC and parental cancer cells H460 (a) and A549 (b) after being treated by fVII-tPDT for 36 J/cm2 635nm laser light, as determined by clonogenic assay. *: p<0.05; **: p<0.01; ***: p<0.001; ****: p<0.0001. c-d. After fVII-tPDT, the CD133+ H460 CSC cells (c) were stained with Annexin V-FITC and then stained with propidium iodide (PI). Untreated CD133+ H460 CSCs were the control cells (d). The cells were photographed under a fluorescent microscope using green (Annexin/FITC for apoptosis), red (PI for necrosis) and bright field channels. Original magnification: 200 x. Modified from Hu et al. Oncotarget 2017 (doi: 10.18632/oncotarget.13644)
Figure 5. TF-targeted PDT (fVII-tPDT) using chemically conjugated fVII-SnCe6 is selective and effective in eradicating angiogenic VEC via binding to TF and by induction of apoptosis and necrosis. a–c. fVII-SnCe6 conjugates retain the binding activity and selectivity to angiogenic VECs (VEGF-stimulated HMVEC: human microvascular endothelial cells), while it has no binding to unstimulated HMVEC. Goat anti-TF was a positive control (a), SnCe6 was separately conjugated with mfVII/Sp (b) and mfVII/NLS (c). d. Representative imaging of crystal violet stained VEGF-stimulated and unstimulated HMVECs right after being treated with fVII-tPDT or ntPDT (2 µM and 635 nm laser light at 36 J/cm2). Control HMVECs include an untreated control and a maximal killing control (completely lysed by 1% Triton X-100). e. Complete eradication (no colonies formed) of angiogenic VEC (HMVEC) by fVII-tPDT using fVII/NLS-SnCe6 or fVII/Sp-SnCe6, whereas ntPDT has no therapeutic effect in killing angiogenic VEC. f. The fVII-tPDT is effective and selective in killing angiogenic VEC, whereas it has no side effects on quiescent VEC (635 nm laser light at 36 J/cm2). Note that the VEC cells without fVII/NLS-SnCe6 (0.0 µM) also served as the light only control as they were also irradiated with 635 nm laser light (36 J/cm2). g. The underlining mechanism of fVII/NLS-tPDT involves rapid induction of apoptosis and necrosis. Annexin V-FITC (green) stains for apoptotic cell membrane (green arrow), while propidium iodide (PI, red) stains for the nuclei of dead cells (red arrows). Modified from Hu et al. Angiogenesis. 2017 (doi: 10.1007/s10456-016-9530-9)
Comment 2. Lines 46-49 describes the primary pathways of PDT’s action. More precisely, type I reactions do yield reactive superoxide species (ROS) from the reaction of oxygen with radicals formed from the electron transfer reaction of excited PSs and biomolecules. Type II reactions categorically produce singlet oxygen through energy transfer from the PS to molecular oxygen.
Response: We appreciate your insightful suggestions. Now the revision reads below.
“Type I reactions produce ROS from the reaction of oxygen with radicals formed from the electron transfer reaction of excited photosensitizers and biomolecules that go on to damage cellular structures and organelles. Type II reactions are when there is a transfer of energy from active photosensitizers to molecular oxygen (3O2), forming singlet oxygen [19].”
Comment 3. How did the authors decide on which proteins overexpressed in cancer cells to feature in the paper?
Response: The decision on which target molecules to feature in this review was based on its expression on cancer cells, or ideally, on multiple tumor compartments, including but not limited to, cancer cells, cancer stem cells and tumor angiogenic vascular endothelial cells. For its common yet selective expression on those tumor compartments, tissue factor was more extensively discussed for the feasibility, efficacy, underlying mechanism and safety of TF-targeted PDT using chemically conjugated fVII-PS (new Figures 2-5).
The inclusion of other targets, most of which are primarily expressed on cancer cell alone, was based on the availability of published bioconjugation methods in targeted PDT. Due to the scope of this review, we apologize if we have unintentionally omitted some of current targets in PDT and/or PDD but will try to include for discussion in future.
Comment 4. Lines 105-108 are repeating ideas presented in the previous statements. They can be removed for clarity.
Response: Thank you for your comments. The repeating sentences have been removed.
“Tissue factor (TF), also known as CD142 [51,52], is the only membrane-bound coagulation factor (factor III) and cofactor and surface receptor for coagulation factor VII (fVII)/activated fVII (fVIIa) [53-55], the latter is a soluble molecule within the coagulation cascade [56]. Under physiological condition, TF is not expressed on peripheral blood lymphocytes (T, B, NK and monocytes) [57-59], nor on quiescent VEC of normal blood vessels in normal tissues and organs [60-62]. TF expression is restricted to the cells that are not in direct contact with the blood, such as pericytes, fibroblasts and smooth muscle cells, which are localized in the sub-endothelial vessel wall and thus sequestered from circulating coagulation fVII. TF is key to initiate blood coagulation and to maintain homeostasis. When vessel wall integrity is damaged, TF in pericytes and smooth muscle cells can be bound by fVII, leaked from circulation, forming TF-fVII complex, which will lead to coagulation cascade activation and eventually clot formation [63].”
Comment 5. For consistency, lines 164-165, 200, 211, 223 may be removed as they are purification methods which are not stated anywhere else in the paper.
Response: We appreciate your comments and suggestions. The purification methods have been removed from lines 164-165, 200, 211 and 223. In addition, we included a few more sentences below to explain the ideas and design of two TF-targeting molecules for chemical bioconjugation of fVII-PS conjugates for TF-targeted PDT.
“The first TF-targeting photosensitizer conjugate was generated by conjugating active-site mutated fVII (K341A) protein to the photosensitizer verteporfin for the PDT treatment of breast cancer in a mouse model [88] and choroidal neovascularization in a rat model [105], the latter is a cause of age-related macular degeneration in humans. The K341A mutation in fVII proteolytic domain significantly reduced its procoagulation activity [65,96,106], while its binding activity to TF retained [96]. The first TF-targeting protein that we constructed for targeted PDT was mfVII/Sp protein [87,88,104], which was composed of murine fVII(K341A), an S peptide (Sp) tag with a mutation at D14N and a polyhistidine tag (His tag, for protein purification and detection) (mfVII(K341A)/Sp(D14 N)/His, abbreviated as fVII/Sp). Later, we made a second TF-targeting protein, fVII/NLS (NLS for nuclear localization sequence) [83,86] with a hope to further bring photosensitizer into nuclear following endocytosis of fVII/TF complex [104]. It is composed of mfVII(K341A) followed by 2 repeats of the wild-type NLS (PKKKRKVG) of SV40 T-antigen and a His tag [83,86]. The fVII proteins were produced by recombinant DNA technology, specifically, by transfecting Chinese hamster ovary cells (CHO-K1, ATCC) with plasmid encoding fVII/Sp and fVII/NLS cDNAs, whereas verteporfin was extracted from liposomal Visudyne (QLT Inc.) via acidification and separation [88].”
Comment 6. Minor changes: In lines 111-112, the phrase “In cancer” may be removed or the sentence can be paraphrased. “Accumulating” should be “Accumulated” and the word “oncotarget” does not exist but as a name of a journal. In line 170, the phrase “in mice” is redundant. In line 228, tumor volume reached 100 “mm3” not “m3”. In line 233, the word “duel” should be “dual”. In line 337, “orthotropic” should be “orthotopic”.
Response: We have removed “In cancer” and changed to “Accumulated” and “target” at lines 111-112. Also, we have cited references here and included two new figures (Figures 2 & 3, also seen above) from our previously published open access articles to support the statement.
“Accumulated evidence from our laboratory and other groups demonstrate that TF is a common yet selective target for cancer cells [64], tumor VECs [65,66] and CSCs [35] (Figures 2 & 3).”
We apologize for the typos in lines 228, 233 and 337. We have made corrections and changes accordingly.
Comment 7. Please discuss more on this topic found in line 395, “PDT is a method of bypassing Pgp MDR currently being researched to improve the efficacy of cancer therapy”.
Response: Thank you for your comments. We have discussed more on targeting P-glycoprotein.
“P-glycoprotein (Pgp) is an efflux transporter found in normal cells, but in cancer cells it contributes to MDR by transporting drugs out of the cell, thus preventing drug action and treatment. Pgp has been found to influence MDR in a variety of cancers, including liver, lung, skin, and more [161-163]. Inhibiting Pgp is of particular interest in oncology in order to overcome MDR [164]. Pgp targeted PDT may be able to bypass drug resistance, as it allows for direct destruction of the tumor promoting transporter [165].”

Reviewer 3 Report
The authors present a review about bioconjugation strategies with photosensitizers for diagnosis and treatment of cancers. They show in a short and clear way an overview of the different techniques to link the photosensittizers to biological components. I list below some tips that in my opinion will improve the visibility and impact of the review.
- Line 2. Please name other different diseases or applications in which PDT or PDD could be also useful.
- Line 41. The difference between PDT and PDD are not clear. Notably, the explanation concerning PPD is confusing. The way how the photosensitazers mark especifically cancer cells/tissues should be clarify it.
- I miss a brief explanation in the introduction about how works PDD since for PDT the authors have positively exposed its mechnanism.
- I suggest to introduce one or two figures which include an overview of the review, showing what is PDT and PDD and the different strategies of bioconjugation of photosensitizers.
- Line 102-108, the text is repetitive and unclear.
- Line 111: “In cancer, accumulating evidence from our laboratory and other groups demonstrate that TF is 112 a common yet selective oncotarget for cancer cells, tumor VECs and CSCs”. References should be added.
- Line 286 a point is missing after more
- The authors show an example with QDs (Line 291) but it is unclear what it is the role of the nanoparticles, are they the emitters and are they bioconjugated? In the introduction, the authors talk about photosensitizers (first, second and third generation) and one assume that they are writing about organic photosensitizers. These should be clarified and to be consistent along the text.
- There are few examples about PDD could the authors improve that?
- Lines 323-328. Review this part. Is it possible PDD without PDT?
- The tittle the review starts with bioconjugation strategies, but after this is the shortest part of the review.
Author Response
Reviewer 3
Comments and Suggestions for Authors
The authors present a review about bioconjugation strategies with photosensitizers for diagnosis and treatment of cancers. They show in a short and clear way an overview of the different techniques to link the photosensitizers to biological components. I list below some tips that in my opinion will improve the visibility and impact of the review.
Comment 1. - Line 2. Please name other different diseases or applications in which PDT or PDD could be also useful.
Response: Thank you for your suggestions and comments. We have briefly mentioned some of other noncancerous diseases and applications of PDT or PDD, as seen below.
“Photodynamic diagnosis (PDD) and therapy (PDT) are emerging, non/minimally invasive techniques for diagnosis and treatment of cancer [1-3]. Besides cancer, PDT has been used to treat age-related macular degeneration [4], microbial infections [5,6], as well as dermatologic, urologic, gynecologic and oral diseases [7-10].”
Comment 2. - Line 41. The difference between PDT and PDD are not clear. Notably, the explanation concerning PPD is confusing. The way how the photosensitizers mark specifically cancer cells/tissues should be clarify it.
Response: We fully agree with you and apologize for the confusion. It has been rewritten, as seen below.
“current PDD usually uses the fluorescence of non-targeted photosensitizer without ROS production, for example, 5-Aminolevulinic acid (5-ALA) with blue or white light excitation [14,15], to identify tumor tissue. The fluorescence will thus hopefully mark pathologic tissue under fluorescent light without destroying it. Thus, PDD, when used as an intraoperative fluorescence-guided diagnosis, may allow for more accurate removal of tumors and lower the risk of unnecessary damage to healthy tissue during surgical operation [16-18].”
Comment 3. - I miss a brief explanation in the introduction about how works PDD since for PDT the authors have positively exposed its mechanism.
Response: Please see our response above under Comment 2.
Comment 4. - I suggest to introduce one or two figures which include an overview of the review, showing what is PDT and PDD and the different strategies of bioconjugation of photosensitizers.
Response: Thank you and Reviewer 2 for your great suggestions. Now we have included one new figure (Figure 1) for easy visualization and elucidation of the ideas and bioconjugation strategies.
Comment 5. - Line 102-108, the text is repetitive and unclear.
Response: Thank you and Reviewer 2 (Comment 4) for sharing the same comments. We have rewritten the whole paragraph, as seen below.
“Tissue factor (TF), also known as CD142 [51,52], is the only membrane-bound coagulation factor (factor III) and cofactor and surface receptor for coagulation factor VII (fVII)/activated fVII (fVIIa) [53-55], the latter is a soluble molecule within the coagulation cascade [56]. Under physiological condition, TF is not expressed on peripheral blood lymphocytes (T, B, NK and monocytes) [57-59], nor on quiescent VEC of normal blood vessels in normal tissues and organs [60-62]. TF expression is restricted to the cells that are not in direct contact with the blood, such as pericytes, fibroblasts and smooth muscle cells, which are localized in the sub-endothelial vessel wall and thus sequestered from circulating coagulation fVII. TF is key to initiate blood coagulation and to maintain homeostasis. When vessel wall integrity is damaged, TF in pericytes and smooth muscle cells can be bound by fVII, leaked from circulation, forming TF-fVII complex, which will lead to coagulation cascade activation and eventually clot formation [63]. ”
Comment 6. - Line 111: “In cancer, accumulating evidence from our laboratory and other groups demonstrate that TF is 112 a common yet selective oncotarget for cancer cells, tumor VECs and CSCs”. References should be added.
Response: Again, thank you and Reviewer 2 for your comments. References have been cited.
“Accumulated evidence from our laboratory and other groups demonstrate that TF is a common yet selective target for cancer cells [64], tumor VECs [65,66] and CSCs [35] (Figures 2 & 3).”
Comment 7. - Line 286 a point is missing after more
Response: We apologized for this error. A period is now added.
“but also function in cytoskeleton organization, intracellular signaling, cell proliferation, and migration [131].”
Comment 8. - The authors show an example with QDs (Line 291) but it is unclear what it is the role of the nanoparticles, are they the emitters and are they bioconjugated? In the introduction, the authors talk about photosensitizers (first, second and third generation) and one assume that they are writing about organic photosensitizers. These should be clarified and to be consistent along the text.
Response: We really appreciate your insightful comments. Indeed, this review focuses on organic and protein photosensitizers. Therefore, we have removed the nanoparticle-QDs study (line 291) from this review and accordingly from Table 1.
Comment 9. - There are few examples about PDD could the authors improve that?
Response: For the non-targeted PDD, we included a few references for PDD in the first paragraph below.
“for example, 5-Aminolevulinic acid (5-ALA) with blue or white light excitation [14,15],”
For currently targeted PDD/PDT, we discussed some near infrared (NIR) dyes, for example, IR700 and IRDye800 CW, which are being used in targeted PDD or PDT. They were included in Table 1 and discussed under 2.2 HER2, 2.6 EGFR and 2.7 P-glycoprotein.
“It is worth noting that near infrared (NIR) dyes are also being developed for cancer imaging and treatment [32]. For those NIR dyes that possess PDD and/or PDT activity, for example, IR700 and IRDye800 CW, we also summarized them briefly in this review (Table 1).”
Comment 10. - Lines 323-328. Review this part. Is it possible PDD without PDT?
Response: Thank you for your concerns. The study discussed in Lines 323-328 was not really PDD. And PDT was not tested. We have rewritten it, as seen below.
“One study examined the specificity in a FRα targeted-photosensitizer bioconjugate on a rat NuTu-19 ovarian adenocarcinoma in a rat peritoneal cavity model. Researchers used 5-(4-Carboxyphenyl)-10,15,20-triphenylporphyrin (Porph) and {N-{2-[2-(2-aminoethoxy)ethoxy]ethyl}[65,69,71] folic acid}-4-carboxyphenylporphyrin (Porph-s-FA) [144]. Four hours after intraperitoneal administering free PS (Porph) and folic acid (FA)-targeted PS (Porph-s-FA), they sacrificed the animals and examined the correlation of FRα expression (by IHC) and cytoplasmic red fluorescence (as an indication of PS under confocal microscopy) and bio-distribution of PS [144]. The results showed a correlation of FRα expression and fluorescence observation in tumor tissues as well as in several normal tissues (ovary and liver) with a tumor to normal tissue ratio of 9.6 (by assaying porphyrins in nanomoles per gram of protein). Another limitation of the study was that PDT was not tested.”
Comment 11. - The tittle the review starts with bioconjugation strategies, but after this is the shortest part of the review.
Response: The review discusses current targets and two bioconjugation strategies (Figure 1) in the use of organic and protein photosensitizers, mainly for PDT with a hop that these approaches in PDT can be adapted for targeted PDD. To reflect it in the title, we have now revised the title of this review to “Current Targets and Bioconjugation Strategies in Photodynamic Diagnosis and Therapy of Cancer”.